# An Overview of Cefiderocol’s Therapeutic Potential and Underlying Resistance Mechanisms

**DOI:** 10.3390/life13071427

**Published:** 2023-06-21

**Authors:** Sara Domingues, Tiago Lima, Maria José Saavedra, Gabriela Jorge Da Silva

**Affiliations:** 1Faculty of Pharmacy, University of Coimbra, 3000-548 Coimbra, Portugal; saradomingues@ff.uc.pt (S.D.); tiago.lima@ff.uc.pt (T.L.); 2Center for Neuroscience and Cell Biology (CNC), University of Coimbra, 3004-504 Coimbra, Portugal; 3CITAB-Inov4Agro, Centre for the Research and Technology of Agro-Environmental and Biological Sciences, University of Trás-os-Montes and Alto Douro, 5000-801 Vila Real, Portugal; saavedra@utad.pt; 4CECAV-AL4AnimalS, Animal and Veterinary Research Center, University of Trás-os-Montes and Alto Douro, 5000-801 Vila Real, Portugal

**Keywords:** cephalosporin, carbapenem-resistant, cefiderocol antibacterial activity, cefiderocol resistance, antimicrobial resistance, healthcare infections, siderophore, iron transporter, ESKAPE

## Abstract

Antimicrobial resistance continues to increase globally and treatment of difficult-to-treat (DTT) infections, mostly associated with carbapenem-resistant (CR) *Pseudomonas aeruginosa*, CR *Acinetobacter baumannii*, and CR- and third-generation-cephalosporins-resistant *Enterobacterales* remains a challenge for the clinician. The recent approval of cefiderocol has broaden the armamentarium for the treatment of patients with DTT infections. Cefiderocol is a siderophore cephalosporin that has shown excellent antibacterial activity, in part due to its innovative way of cell permeation. It is relatively stable compared to most commonly found carbapenamases. However, some resistant mechanisms to cefiderocol have already been identified and reduced susceptibility has developed during patient treatment, highlighting that the clinical use of cefiderocol must be rational. In this review, we summarize the current available treatments against the former resistant bacteria, and we revise and discuss the mechanism of action of cefiderocol, underlying the biological function of siderophores, the therapeutic potential of cefiderocol, and the mechanisms of resistance reported so far.

## 1. Introduction

The discovery of antibiotics in the 20th century was a milestone in the history of medicine and healthcare. However, bacteria are constantly evolving, and have been developing resistance mechanisms against virtually all available antibiotics [1]. In order to uniformize the definitions associated with multi-resistance, a joint initiative from the European Centre for Disease Prevention and Control and the Centers for Disease Control and Prevention defined the resistance profile of bacteria classified as multidrug-resistant (MDR), extensively drug-resistant (XDR), and pandrug-resistant (PDR); the standardized definitions and associated resistance profiles were elaborated for bacteria highly prone to develop resistance and frequently responsible for infections in the healthcare system [2].

After the golden age, the rhythm of antibiotic discovery has decreased, with fewer new antibiotics being introduced into the clinical practice over the last few decades, resulting in the current antibiotic crisis, where there are very scarce options to treat multidrug-resistant bacterial infections, especially the ones related to Gram-negative bacteria [3]. MDR is associated with an increased risk of mortality [4] and it is predictable that the lack of therapeutic options will lead to 10 million deaths in 2050 if new antibiotics are not developed [5].

The World Health Organization (WHO) has listed the pathogens that urgently need new antibiotics, with MDR Gram-negative bacteria included in the critical priority level of the list [6,7]. In the last five years, five antibiotics active against MDR Gram-negative bacteria were approved; however, they are all modified agents of already known classes [8], which increases the possibility for antibiotic resistance emergence.

This review will focus on the latest available therapy for pathogens classified at the critical priority level of the WHO list, the carbapenem-resistant *Acinetobacter baumannii* and *Pseudomonas aeruginosa,* and carbapenem- and third-generation-cephalosporins-resistant *Enterobacterales*, and on the novel cephalosporin cefiderocol, which offers a novel approach to cell permeation and shows promise as a treatment option for these challenging infections. The mechanism of action and therapeutic guidelines of cefiderocol will be analyzed, along with a discussion on the potential emergence of resistance based on recent in vitro and in vivo studies.

## 2. Top-Priority Pathogens and Current Last-Line Therapeutic Options

### 2.1. Carbapenem-Resistant Acinetobacter baumannii

The microorganism that is at the top of the priority list is carbapenem-resistant *A. baumannii* (CRAB); resistance to carbapenems is usually associated with resistance to several other antibiotics theoretically active against *A. baumannii* [9]. This Gram-negative coccobacillus, which was not particularly known to be a highly pathogenic bacterium, has emerged in recent years as the cause of several nosocomial infections worldwide, including ventilator-associated pneumonia and bloodstream and wound infections (WHO 2017), especially due to its ability to develop antibiotic resistance [10] and resist desiccation [11], persisting in the hospital environment for long periods of time. Besides the intrinsic resistance to several classes of antibiotics, *A. baumannii* can easily acquire resistance via horizontal gene transfer events [12,13].

Different resistance mechanisms preclude the general use of β-lactams, sulbactam, aminoglycosides, and fluoroquinolones usually active against susceptible strains of *A. baumannii* [9]. According to European and American guidelines, the agent of choice for treatment of CRAB infections is ampicillin-sulbactam; alternatives include minocycline, tigecycline, polymyxins, or cefiderocol [4,9]. Combination therapy is recommended for moderate to severe infections, including high-dose ampicillin-sulbactam plus at least one agent with in vitro activity among minocycline, tigecycline, polymyxin B, extended-infusion meropenem, or cefiderocol [9].

### 2.2. Carbapenem-Resistant Pseudomonas aeruginosa

*P. aeruginosa* is an opportunistic nosocomial pathogen mainly responsible for infections in patients with a compromised immune system, especially pneumonia and other respiratory infections, as well as bloodstream and wound infections [6,14]. Carbapenem resistance is linked with increased mortality in bloodstream infections [6]. *P. aeruginosa* is commonly found in the hospital environment due to its presence in water sources and medical devices [14].

MDR *P. aeruginosa* has a multiplicity of resistance mechanisms that prevent the use of antibiotics that should be active against this species, namely penicillins, cephalosporins, fluoroquinolones, aminoglycosides, and carbapenems; the term difficult-to-treat resistance (DTR) has been proposed for strains with non-susceptibility to piperacillin-tazobactam, ceftazidime, cefepime, aztreonam, meropenem, imipenem-cilastatin, ciprofloxacin, and levofloxacin [9].

Overall, high-certainty evidence to advise on the adequate therapy for carbapenem-resistant *P. aeruginosa* (CRPA) is lacking. For non-severe or low-risk CRPA, monotherapy with old antibiotics, such as polymyxins and aminoglycosides, are recommended. The recommendation for DTR-CRPA severe infections is the use of ceftolozane-tazobactam; two drugs should be combined if using polymixins, aminoglycosides, or fosfomycin [4]. Due to the lack of evidence, the use of new β-lactam agents, including ceftazidime-avibactam or cefiderocol, are not recommended by European guidelines [4], but are advised by American recommendations [15].

### 2.3. Carbapenem- and Third-Generation-Cephalosporins-Resistant Enterobacterales

*Enterobacterales* include Gram-negative rod bacteria that inhabit in a variety of environments; numerous species are part of the human and animal microbiota, being widely release to the environment. As opportunistic pathogens, they are also implicated in several bacterial infections, including urinary tract infections and bloodstream and ventilator-associated pneumonia [6,16,17]. Extended-spectrum β-lactamase (ESBL) or AmpC β-lactamase production is associated with strains resistant to third generation cephalosporins; the rise of this type of resistance led to the increased use of carbapenems in empirical treatments, with a consequent emergence of carbapenem-resistant *Enterobacterales* (CRE), mainly associated with carbapenemases production [6,18].

The treatment recommendations for infections due to third-generation-cephalosporin-resistant *Enterobacterales* include carbapenems for bloodstream and severe infections, piperacillin-tazobactam, amoxicillin/clavulanic acid or quinolones for low-risk and non-severe infections, cotrimoxazole for non-severe complicated urinary tract infections (cUTI), and intravenous fosfomycin for cUTI. After the stabilization of patients, the carbapenem should be replaced by classical β-lactam/β-lactamase inhibitors, quinolones, or cotrimoxazole [4]. The American guidelines recommend different antibiotics based on the genotypic mechanism of resistance. Infections due to *Enterobacterales* that produce ESBL can be treated with nitrofurantoin or cotrimoxazole (uncomplicated cystitis); ertapenem, meropenem, imipenem-cilastatin, ciprofloxacin, levofloxacin, or cotrimoxazole (pyelonephritis or cUTI); and carbapenem followed by oral fluoroquinolone or cotrimoxazole after initial clinical response (infections outside the urinary tract) [15]. When resistance to third-generation cephalosporins is present in strains with a moderate to high risk of AmpC expression due to an inducible *ampC*, cefepime is recommended if the minimum inhibitory concentration (MIC) is ≤2 µg/mL or a carbapenem when the cefepime MIC ≥ 4 µg/mL; ceftriaxone, nitrofurantoin, cotrimoxazole, or single-dose aminoglycoside for uncomplicated cystitis; and cotrimoxazole or fluoroquinolones for invasive infections [9].

CRE severe infections, outside the urinary tract, should be treated with meropenem-vaborbactam, ceftazidime-avibactam, or imipenem-cilastatin-relebactam [4,15]; if susceptibility to meropenem is present, extended infusion is recommended [15]. If a metallo-β-lactamase is present or the strains are resistant to all other antibiotics, cefiderocol or ceftazidime-avibactam in combination with aztreonam may be used. Non-severe infections can be treated with old antibiotics, such as polymyxins or tigecycline as alternatives to β-lactams. cUTI can be treated with aminoglycosides, ciprofloxacin, levofloxacin, or cotrimoxazole and pneumonia with tigecycline [4,15]. Ciprofloxacin, levofloxacin, cotrimoxazole, nitrofurantoin, or single-dose aminoglycoside can be employed in the treatment of uncomplicated cystitis [15].

## 3. Cephalosporins

The β-lactam antibiotics are one of the most commonly prescribed antibiotics in many countries. Among these, cephalosporins are used to treat infections such as pneumonia, urinary tract infections, and skin and soft tissue infections. Cephalosporins are classified as β-lactam antibiotics, originally derived from the fungus *Cephalosporium* sp., known today as *Acremonium* sp. [19].

They act in bacteria by binding to the penicillin binding proteins (PBPs), enzymes located in the cytoplasmatic membrane that cross-link the peptidoglycan units, inhibiting the building of the cell wall and leading to cell death. Cephalosporins are used for the treatment of various infections caused by both Gram-positive and Gram-negative bacteria. They can be administered orally or parenterally, and are used in skin infections, pneumonia, meningitis, and infections caused by bacteria resistant to others antibiotics. So far, they are classified into five generations, according to their antibacterial spectrum of action and their temporal release.

The first cephalosporins launched, the so-called first-generation cephalosporins, act mainly on Gram-positive bacteria, while the following generations progressively increased their effectiveness against Gram-negative bacteria; the third and fourth generations were more powerful against resistant Gram-negative bacteria and lost some effectiveness against Gram-positive bacteria. Fifth-generation cephalosporins have a broad bacterial spectrum and act on Gram-positive and Gram-negative bacteria. Some fifth-generation cephalosporins stand out for their activity against Gram-positive bacteria, in particular methicillin-resistant *Staphylococcus aureus* (MRSA), such as ceftaroline. Others, like the combinations of ceftazidime/avibactam and ceftolozane/tazobactam, have been developed to address most difficult-to-treat Gram-negative infections [20].

The emergence and dissemination of ESBL producers, mostly among *Enterobacterales*, led to the use of last-resort antibiotics, such as carbapenems. In the last few years, a blast of infections caused by carbapenem-resistant Gram-negative bacteria has been observed, in part due to the production of carbapenemases, like NDM type, OXA-48-like, and KPC type, by clinical strains. Colistin has been re-introduced into the clinical practice to fight these MDR infections, but its toxicity and the emergence of colistin-resistant strains are factors to consider in its therapeutic use [6,21,22,23,24,25].

The outer membrane (OM) functions as a permeability barrier in Gram-negative bacteria, slowing down the passive diffusion of hydrophobic compounds of high molecular weight into the cell, which are active in Gram-positive bacteria, and letting only small hydrophilic molecules penetrate through the porin channels. In addition, bacteria can have active transporters that efflux the antibiotics from the periplasmic space, preventing the β-lactam antibiotics from reaching the target, the PBPs, located in the cytoplasmic membrane [26].

Different strategies have been explored to increase the antibiotic concentration in the periplasmic space, namely inhibition of efflux pumps, use of OM disrupters to increase permeability, modifications of the antibiotic structure or electrical charge, and developing antibiotics that cross the OM using iron transporters—the siderophores [27,28,29,30,31].

## 4. Siderophores: Biological Function

Virulence factors are cellular structures, molecules, and regulatory systems that enable bacteria to colonize the host at the cellular level and promote the development of disease. These factors can be secretory, membrane/cell-wall-associated, and cytosolic in nature, and they include adherence and invasive factors, exotoxins, endotoxin in the case of Gram-negative bacteria, capsules, and other surface components. There is a lot of interest in these traits that can be disrupted and used for therapeutic purposes.

Iron is an essential nutrient for both host and microbial cells. Free iron levels are extremely low in the host since this metal is largely bound to proteins, and much more limited during the process of infection, a process known as nutritional immunity [32].

In general, animals have developed ways of retaining iron from body fluids, with the main objectives of preventing bacterial development and maintaining iron homeostasis. The blood therefore transports most of the iron in the form of molecular complexes, not being freely accessible to bacteria, but rather, and mainly, bound to intracellular haemoglobin or extracellular proteins such as transferrin, which has a high affinity to iron under physiological conditions. During infection, organic acids are produced, which reduce the pH, and Fe^3+^ is released into the medium. To prevent the obtention of iron by bacteria, neutrophils synthesize lactoferrin that shows a much higher affinity for ferric ions in acidic environment than transferrin [33].

Bacteria require intracellular iron for their metabolism, which they achieve by producing siderophores, molecules with a high affinity for binding iron. Siderophores are usually small organic molecules that are secreted into the environment and effectively compete with host proteins to bind scarce free iron (Fe^3+^). Bacteria produce different siderophores, depending on the species, such as enterobactin, pioverdin, and salmochelin. Due to the presence of functional groups, these molecules form strong complexes with iron ions. OM receptors recognize these siderophore–iron complexes and transport them to the periplasmic space with the assistance of protein complexes in the cytoplasmic membrane (TonB and ExbB/ExbD family) that generate the necessary energy for active transportation. The iron is then incorporated into the respiratory chain once it is inside the bacteria cytoplasm [34].

## 5. Cefiderocol

### 5.1. Mechanism of Action

The chemical structure of cefiderocol and unique mechanism of entry into the bacterial cell contribute to its potent antibacterial activity and ability to overcome resistance [35].

The molecular structure of cefiderocol has similarities with both cefepime and ceftazidime (Figure 1), sharing with the former a pyrrolidin group in the C3 side chain, which promotes the stability of the molecule against the action of β-lactamases, improving the antibacterial activity, while with the second, it shares a carboxy-propanoxymino group in the side chain at C7, which improves transport through the bacterial outer membrane [36].

Unlike cefepime and ceftazidime, cefiderocol has a chlorocathecol residue at the end of the C3 chain, which makes it a siderophore. Natural siderophores, such as enterobactin from *Escherichia coli* and pyoverdin from *P. aeruginosa*, also show a cathecol group as an iron chelator. This cathecol group is recognized by the iron active transporters located in the bacterial OM, allowing the antibiotic to pass through the membrane and to reach the periplasmic space where cefiderocol bind to the PBPs, primarily PBP-3. This promotes a faster and higher concentration of the antibiotic in the periplasm, in addition to the passive diffusion through porins, the traditional passage used by β-lactams (Figure 2).

This innovative strategy has been called “Trojan horse”, which gave this siderophore–cephalosporin the commercial name of “Fetroja”, discovered and developed by Shionogi & Co., Ltd., Osaka, Japan. This strategy overcomes the resistance mechanism of loss of porins. Moreover, the chemical structure of cefiderocol renders this antibiotic more stable against a variety of Ambler A, B, C, and D beta-lactamases, namely KPC and ESBLs from class A, AmpC, the carbapenemase OXA-48 from class D serine β-lactamases, and the metallo-β-lactamases NDM, VIM, and IMP [37].

Recently, it was also demonstrated that cefiderocol remained completely active against minor carbepenemases (SME, NmcA, FRI, and IMI types) produced by *Enterobacterales* [38].

### 5.2. Therapeutic Indications

Cefiderocol was recently released in the market of the United States of America (USA) and Europe and approved by the Food and Drug Administration (FDA) in 2019 [39] and European Medicines Agency (EMA) in 2020 [40], respectively.

Due to its innovative cell entry mechanism and ability to overcome some forms of bacterial resistance, cefiderocol is a novel cephalosporin focused on the treatment of various infections caused by MDR Gram-negative bacteria [35].

Its chemical structure, especially the cathecol residue on the side chain at position 3, provides its superior activity (Figure 1). Cefiderocol has demonstrated activity against members of the *Enterobacteriaceae* family, such as *E. coli* and *Klebsiella pneumoniae*, and non-fermenter bacilli *P. aeruginosa* and *A. baumannii*. However, it is not effective against Gram-positive bacteria like *S. aureus* or *Streptococcus pneumoniae* [41,42].

In addition to being highly stable against a broad range of β-lactamases, including ESBLs, AmpC, and carbapenemases, cefiderocol has demonstrated identical or superior activity against aerobic Gram-negative bacilli compared to ceftazidime-avibactam and meropenem, including MDR *A. baumannii* and *K. penumoniae* carbapenemase (KPC)-producing *Enterobacterales*. Furthermore, it showed higher potency than ceftazidime-avibactam against resistant phenotypes of *P. aeruginosa* and *Stenotrophomonas maltophilia* [43].

In Europe, it is recommended to treat infections in adults caused by Gram-negative aerobic organisms with limited therapeutic alternatives, always considering the general rules for the rational use of antibacterial agents [40]. In the case of the USA, the therapeutic indications established for cefiderocol by the respective regulatory agency, the FDA, are also only applicable to infections in adults who have reduced or no alternative therapeutic. The FDA has approved cefiderocol for the treatment of cUTI, including pyelonephritis and hospital-acquired bacterial pneumonia (HABP), including ventilator-associated pneumonia caused by susceptible Gram-negative bacteria in patients 18 years of age or older, always bearing in mind the rational use of antimicrobial drugs [39].

However, in view of the ongoing scientific knowledge, it is expected that this drug will be used for other infections by resistant strains in an “off-label” way, especially due to its effectiveness combined with a good safety profile and current low resistance potential. The performance of the APEKS-cUTI, APEKS-NP, and CREDIBLE-CR clinical trials contributed to this optimism around cefiderocol. The APEKS-NP study [44] was a Phase 3 multicenter study that evaluated the efficacy and safety of cefiderocol compared to the best available therapy (BAT), meropenem, in the treatment of HABP. The results showed that cefiderocol was not inferior to BAT, achieving the primary endpoint of clinical cure at day 14, also demonstrating a favorable safety profile compared to BAT. Similar results were obtained with the APEKS-cUTI [45] when comparing cefiderocol with the combination imipenem–cilastatin in the treatment of cUTI. CREDIBLE-CR [46] was a trial that involved cefiderocol and the BAT for the treatment of serious infections caused by carbapenem-resistant Gram-negative bacteria (CRGNB). Cefiderocol exhibited activity against CRE and CRPA. The Infectious Diseases Society of America has published guidelines, focused especially on *Enterobacterales* ESBL producers, in CRE and CR-PA, recommending that in pyelonephritis and cUTI caused by CRE or CRPA, cefiderocol could be used as the preferential treatment along with some new combinations of β- lactams and β-lactamase inhibitors. However, in CRE or CRPA infections outside the urinary tract, it is recommended to save cefiderocol for situations of resistance to other therapeutic options.

Nonetheless, very recent reports of cefiderocol-based regimens suggest that the guidelines and evidence in the treatment of CRAB infections are conflicting. The outcome and safety profile did not differ significantly from the colistin regimen given to the patients [47,48].

### 5.3. Mechanisms of Resistance

The emergence of carbapenem-resistant infections has been increasing in the last years, and novel antibiotics are sorely needed. Cefiderocol appears as a promising new antibacterial with an innovative way of entering the cell periplasmic space. Despite its similar structure with ceftazidime and cefepime (third- and fourth-generation cephalosporins, respectively), cefiderocol shows an increased stability to various β-lactamases (Figure 1), including AmpC and ESBL [43].

Taking into consideration its excellent in vitro activity and the results of Phase 3 clinical trials, it would be expected to have remarkable in vivo activity. Indeed, in recent years, and with the clinical approval of cefiderocol in diverse countries, it has been shown that this siderophore–cephalosporin has been effective in MDR infections, with a higher activity than ceftazidime–avibactam and meropenem [43]. Nonetheless, others reports point to a similar activity and safety profile to colistin in the treatment of CRAB infections [47]. More real-world studies are needed to settle accurate guidelines in the clinical picture. In the meantime, in vitro resistance of some species to cefiderocol has been reported, even in clinical isolates collected in countries like China, where cefiderocol is not yet approved for clinical use [49,50]. Table 1 shows examples of studies that demonstrate the clinical efficacy of cefiderocol and resistance already identified in diverse bacterial species. As EUCAST and CLSI have different MIC breakpoints for cefiderocol, >2 mg/L [51] and 8 mg/L [52], respectively, resistance was considered based on the European breakpoints.

The resistance mechanisms that have been reported so far fall in four categories are the most frequently reported resistance mechanisms against β-lactams. The expression of the metallo-β-lactamase NDM and β-lactamases PER and VEB have been associated with a reduction in cefiderocol susceptibility [61,62,68,70,78,79,80,81]. In *A. baumannii*, the conjugation of cefiderocol with avibactam led to a decreased MIC of the former, suggesting that expression of β-lactamases in this species is linked with resistance [82]. Secondly, structural changes in the β-lactamases AmpC and KPC confer reduced susceptibility to cefiderocol [83,84,85]. Thirdly, mutations in the target gene of cefiderocol, the *pbp3* gene, might contribute to resistance [63,86,87,88]. Lastly, and due to its unique way of cell permeability, it is unsurprising that a reduced expression or mutation of genes involving iron transport pathways, especially the siderophore receptor genes (e.g., *pirA, cirA, ton, piuA*), are associated with cefiderocol resistance in different bacterial species [61,74,77,89,90,91,92,93,94].

The majority of these studies were conducted in vitro since many are merely intuitive, like the modification of the target, reduced expression, or mutations in the genes of the iron transporters. Indeed, some strains, such as those in *Klebsiella* spp., show these mutations without ever having been exposed to cefiderocol before. Also, the development of resistance during treatment with cefiderocol has been reported. Multiple copies of *bla*_NDM_ genes have been found and associated with resistance in *Enterobacterales* through translocation events [95,96]. Increased MICs to cefiderocol were also observed during treatment in a patient with a *P. aeruginosa* infection. The isolates collected during infection time showed variable and gradually increasing levels of resistance to all β-lactams. Unexpectedly, not only one mechanism, most likely the production of β-lactamases, but diverse mechanisms were linked to the resistance to cefiderocol, namely mutations in iron transporter proteins, overexpression of MexAB-OprM efflux pump, and overproduction of *ampC* gene [76,95]. Wide substrate efflux pumps have also been implicated in the reduction of cefiderocol susceptibility. AxyABM overproduction, a resistance nodulation division (RND) efflux system, was associated with resistance to ciprofloxacin, ceftazidime, and meropenem, and increased the MIC of cefiderocol in *Achromobacter* spp. [58].

The introduction of cefiderocol in clinical therapeutics is still recent. While many studies point to an excellent efficacy to treat carbapenem-resistant and MDR infections, others highlight the potential for resistance development [97]. Recently, clinical carbapenem-resistant *Enterobacterales* isolates showing an increased MIC to cefiderocol (MIC > 4 mg/L) were never exposed to this cephalosporin, and were compared to wild-type strains and species-specific reference genomes to understand the putative mechanisms responsible for the high MICs. The findings suggested that an individual antimicrobial resistance marker was not consistent to define a resistance base line to cefiderocol that is linked to the co-expression of different β-lactamases (e.g., carbapenemase and AmpC) along with permeability defects [98]. Clinical isolates have demonstrated this trend, with multiple resistance mechanisms identified rather than a single specific mechanism [99].

Heteroresitance can also have a significant impact on the clinical outcome, as it results from the emergence of resistant subpopulations during treatment, leading to therapeutic failure and the spread of resistant strains. The prevalence of heteroresistance to cefiderocol has been suggested as a reason for the suboptimal effectiveness of cefiderocol in fighting carbapenem-resistant bacteria, especially *A. baumannii* [100,101].

## 6. Conclusions

During the last few decades, Gram-negative bacterial resistance to antibiotics has been increasing at a global level, especially carbapenem-resistant bacteria, constituting a current and future public health issue that must be addressed. A collaborative approach involving both researchers and pharmaceutical industries has recently provided novel promising antimicrobials that have broadened the therapeutic armamentarium. Clinical trials and some case reports have shown that cefiderocol is an effective treatment for the majority of MDR Gram-negative infections, including carbapenem-resistant *Enterobacterales*, *P. aeruginosa,* and *A. baumannii*. Its excellent antibacterial activity mostly lies in its unique way of penetrating the bacterial cell, quickly reaching the targets located in the outer leaflet of the cytoplasmic membrane. Its stereochemical structure also allows a relative stability against hydrolysis for most carbapenemases, namely the most frequently identified (KPC, NDM, VIM, IMP, and OXA-carbapenemases). Nevertheless, resistance to cefiderocol has already been reported, even in strains never before exposed to this antibiotic, due to alterations in the iron transporters. A few clinical cases showed development of resistance in vivo during patient treatment with cefiderocol, which is a huge concern. The studies reporting reduced susceptibility to cefiderocol suggest that in clinical strains more than one mechanism of resistance is involved. It should be taken in consideration that EUCAST firstly recommended that the antimicrobial susceptibility testing to cefiderocol could be performed using broth microdilution or disk diffusion; in the former case, iron-depleted Mueller–Hinton broth should be employed [102]. However, due to limitations in MIC determination with commercially available tests, the current recommendations of EUCAST are the determination of the antimicrobial susceptibility to cefiderocol using disk diffusion (https://www.eucast.org/ast-of-bacteria/warnings accessed on 14 June 2023). Additionally, clinicians should be aware that cefiderocol has strict therapeutic guidelines and it is only recommended in cUTI and HABP infections caused by bacteria that are already MDR and have underlying multiple resistance mechanisms. Also, the expression of multiple copies of β-lactamase appears to affect its efficient activity. Apparently, the association of cefiderocol with a new β-lactamase inhibitor could be a promising strategy, as suggested in a study with *A. baumannii* infections. Indeed, this pathogen remains quite challenging, with some case reports showing conflict on the efficacy of cefiderocol versus other antibiotics. So far, cefiderocol must be used with caution, following the guidelines of regulatory agencies and evaluating the need of patient therapy that requires the knowledge of the pathogen species and antibiogram, prior colonization, previous antibiotic therapy failure, and local of infection to optimize antibiotic prescription. Many more studies are required to establish a clinically efficient profile for cefiderocol and to understand the potential of resistance emergence.

## Figures and Tables

**Figure 1 life-13-01427-f001:**
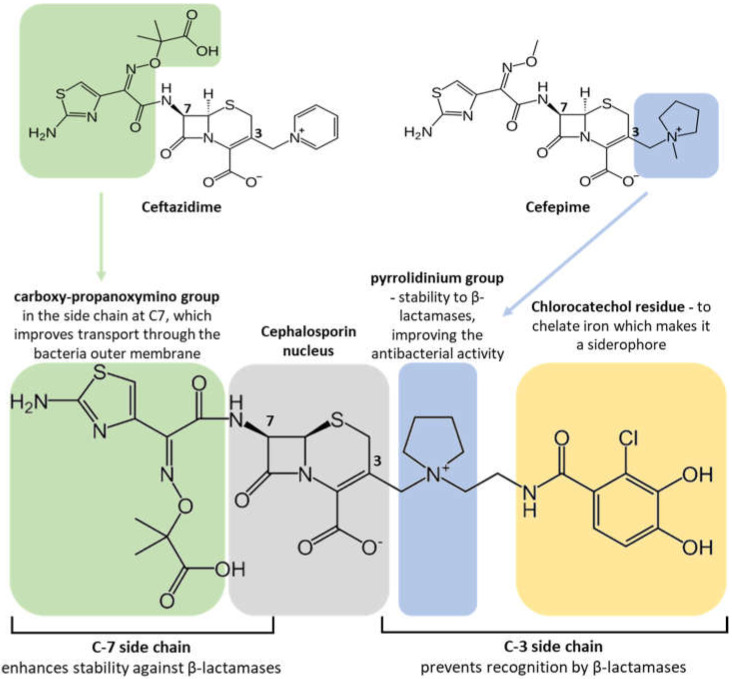
Chemical structure of cefiderocol.

**Figure 2 life-13-01427-f002:**
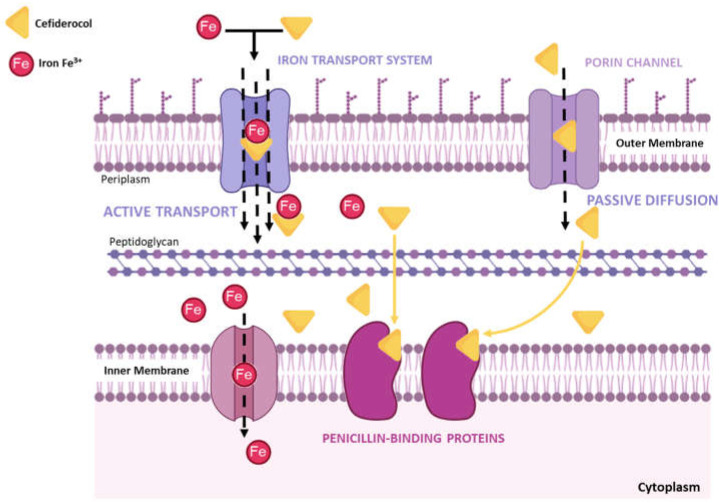
Mechanism of action of cefiderocol.

**Table 1 life-13-01427-t001:** Antibacterial activity of cefiderocol against diverse Gram-negative bacteria and resistance or reduced susceptibility identified in these studies.

Bacteria	Susceptibility		Resistance	
*Achromobacter* spp.	Cefiderocol appeared as a promising therapeutic alternative for managing *Achromobacter* infections in patients with cystic fibrosis.	[53,54,55,56,57]	Overexpression of AxyABM efflux pump in *A. xylosoxidans* was associated with a threefold higher cefiderocol MIC.	[58]
*Acinetobacter baumannii* Complex	SIDERO-WT study 2014 reports 100% of susceptibility of 158 *A. baumannii* complex isolates. SIDERO-WT studies from 2015 to 2019 report percentages of susceptibility to cefiderocol ranging 97.6% to 99.1%.	[59,60]	Twenty-one cefiderocol-non-susceptible carbapenem-resistant *A. baumannii* isolates were characterized, highlighting the contribution β-lactamases, including the presence of an ESBL (PER-1), and deficiency of the iron siderophore transporter PiuA in several isolates. In the second study, by investigating a series of *A. baumannii* clinical isolates with elevated MICs of cefiderocol, the authors showed that PER-like β-lactamases and, to a lesser extent, NDM-like β-lactamases, significantly contributed to reduced susceptibility to cefiderocol. Thirdly, cefiderocol resistance was associated with reduced expression of the siderophore receptor gene *pirA* in *A. baumannii* isolates.	[61,62,63]
*Burkholderia cepacia* Complex	Cefiderocol was more potent in vitro than cefepime, ceftazidime-avibactam, ceftolozane-tazobactam, ciprofloxacin, and colistin.	[54,59,64,65]	Only 1 of the 4, 1 of the 12, and 5 of the 89 isolates tested had a cefiderocol MICs of 16, 8, and ≥8 μg/mL, respectively.	[59,64,65]
*Citrobacter freundii* Complex	All the isolates tested in the SIDERO-WT study 2014 (n = 303) were susceptible to cefiderocol.	[59,66]	Only 1 of the 32 and 1 of the 252 isolates tested had a cefiderocol MIC of 8 μg/mL.	[64,65]
*Citrobacter koseri*	SIDERO-WT study 2014 involving 73 isolates of *C. koseri* with MICs ranging from 0.006 to 4 μg/mL.	[59]	Study involving 73 isolates of *C. koseri* with MICs ranging from 0.008 to 4 μg/mL. In the second study, 1 of 169 isolates had a MIC of 8 μg/mL.	[59,65]
*Escherichia coli*	SIDERO-WT study 2014 involving 1529 isolates of *E. coli* with MICs ranging from ≤0.002 to 4 μg/mL. The MIC_90_ value of cefiderocol against *E. coli* isolates was 0.5 and 1 μg/mL in the second and third study, respectively.	[59,65,66]	In total, 10 out of 142 *E. coli* isolates were resistant to cefiderocol. In 26 of 1158 *E. coli* isolates harboring NDM-5 high levels of cefiderocol resistance was reported, in the second study. In the third study, a multidrug-resistant ST167 *Escherichia coli* clinical isolate recovered from a patient hospitalized in Switzerland produced NDM-35 showing ca. 10-fold increased hydrolytic activity toward cefiderocol compared to NDM-1.	[49,67,68]
*Enterobacter cloacae* Complex	In SIDERO-WT-2014 study and in another study involving 514 and 103 isolates of *E. cloacae*, respectively, the MIC_90_ value was 1 μg/mL.	[59,66]	In the first study, the authors report 2 cefiderocol-resistant ECC isolates in a collection of 10 isolates collected from diabetic patients. In the second study, the potential role of the VIM-1 carbapenemase in cefiderocol resistance in the ECC was highlighted. This effect is probably enhanced due to combination with additional mechanisms, such as ESBL production and siderophore inactivation. The presence of the NDM β-lactamase facilitates the emergence of resistance via nonsynonymous mutations of the c*irA* catecholate siderophore receptor in the third study.	[67,69,70]
*Klebsiella (Enterobacter) aerogenes*	The MIC_90_ value of cefiderocol against *E. aerogenes* isolates in was 0.5 μg/mL in both studies with 238 and 100 isolates.	[59,66]	In this study, 1158 cefiderocol resistant isolates were identified, of which 20 (1.7%) were *K. aerogenes.*	[49]
*Klebsiella pneumoniae*	The MIC_90_ value of cefiderocol against 765 and 100 *K. pneumoniae* isolates was 0.5 and 0.125 μg/mL, in the first and in the second study, respectively. In the second study MIC values ranged between ≤0.063–2 μg/mL.	[59,66]	In the first study, 7 out of 91 *K. pneumoniae* were resistant to cefiderocol. In the second study, 1158 cefiderocol-resistant isolates were identified, of which 798 (68.9%) were *K. pneumoniae*. In the third study, the authors characterized four cefiderocol-non-susceptible *K. pneumoniae* strains (4/86, 4.7%).	[49,50,67]
*Klebsiella oxytoca*	In the SIDERO-WT-2014 (505 isolates) and 2015 (349 isolates) the MIC values of cefiderocol ranged between ≤0.002 and 2 μg/mL and MIC_90_ value was 0.25 and 0.5 μg/mL, respectively.	[59,65]	In this study, 1158 cefiderocol resistant isolates were identified, of which 23 (2%) were *K. oxytoca.*	[49]
*Morganella morganii*	All of 37 and 32 isolates tested were classified as susceptible to cefiderocol, respectively, in two studies.	[71,72]	Only 1 out of 1158 isolates of *M. morganii* was classified as cefiderocol-resistant in this study.	[49]
*Proteus* spp.	All of 89 isolates tested were classified as susceptible to cefiderocol.	[71]	Only 2 out of 52 isolates were resistant to cefiderocol. In a second study, 1 of 10 isolates resistant to carbapenems was resistant to cefiderocol.	[67,72]
*Providencia rettgeri*	Treatment of complicated urinary tract infections (cUTI), due to Gram-negative bacteria in patients with limited or no alternative treatment options	[16]	One strain was obtained from the blood of a patient and it was resistant to all antimicrobials tested including the cefiderocol	[16,73]
*Pseudomonas aeruginosa*	In both studies all the collection of 120 and 33 isolates, respectively, were susceptible to cefiderocol. Five consecutive annual SIDERO-WT Studies reports a 99.9% of susceptibility to cefiderocol in a total of 7700 isolates.	[54,60,74]	Whole genome sequencing of *P. aeruginosa* non-susceptible to cefiderocol identified mutations in major iron transport pathways. The second study reports in vivo development of cefiderocol resistance among four sequential *P. aeruginosa* clinical isolates ST244 recovered from a single patient, without exposure to cefiderocol.	[75,76]
*Serratia* spp./*Serratia marcescens*	In SIDERO-WT-2014 study, in 503 isolates of *Serratia* spp. MIC_90_ value was 0.25 μg/mL and in another study involving 103 isolates of *S. marcescens* the MIC_90_ value was ≤0.0063 μg/mL.	[59,66]	In total, 14 out of 1158 isolates of *S. marcescens* were classified as cefiderocol-resistant in this study.	[49]
*Stenotrophomonas maltophilia*	Twenty-five meropenem-resistant *S. maltophilia* were susceptible to cefiderocol. In the second study, all the 7 isolates tested were also susceptible. SIDERO WT studies of 2014 and 2017 reports 100% of susceptibility to cefiderocol in 21 and 187 isolates of *S. maltophilia*, respectively.	[54,60,74]	*S. maltophilia* strains evolved cefiderocol resistance through different genetic pathways, but often involved iron transport.	[77]

## Data Availability

Not applicable.

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
