# Peer review of "An Overview of Cefiderocol’s Therapeutic Potential and Underlying Resistance Mechanisms"

_life, 2023, doi:10.3390/life13071427_

Round 1

Reviewer 1 Report

Authors summarized the antibiotics commonly used in practice concerning CRE, CRAB and CR-Pseudomonas. 

They review the potential interest of cefiderocol, a new antibiotic, using siderophore with a novel mechanism of resistance using iron transport system.

They also, review the literature concerning the resistance described up to this date.

Some suggestions:

1) in the abstract :

the use of "and CR and 3rd Gen cephalosporin" is hard to read (including in throughout the manuscript). I would prefer using the term "carbapenem-producing Enterobacterales or ESBL-producers

2) Introduction: Line 43 after citation [3]:

You may add "especially, the ones related to gram-negative bacteria"

3) Page 2 line 53 same suggestion as for the abstract concerning the term "CRE"

4) Page 3 lines 110-130 : for me it is out of the scope of the article, at least it should be shortened. Indeed, you may need to open the discussion about the trend of a "carbapenem-sparing era", and the use / interest of new molecules including ceftazidim avibactam, and carbapenem + inhibitors, and why they do have interest right now.

5) Page 7 Line 295 after citation [47]. You should add a word about the Kaye S. article in the NEJM which shows similar outcome for colistin monotherapy or in combination against CRAB (78%), in pneumonia (70%); 43% vs 37% mortality. https://evidence.nejm.org/doi/abs/10.1056/EVIDoa2200131

6) In the discussion page 10 , you may clarify the point that previously we were supposed to test Cefidericol using broth dilution for MIC determiniation with or without iron; currently EUCAST is calibrated using standard E-test manufactered by Shinogi.

7) Conclusion : I would definitely conclude towards cefiderocol insterest in comparison to new BL+inhibitors, in case other regimen are compromised; which is not clearly stated in the present manuscript.

Author Response

We would like to express to the reviewers our sincere gratitude for the time and effort dedicated to reviewing the manuscript and your valuable and insightful and constructive comments to improve the overall quality of the manuscript.

Below you can find the answers to the reviewers.

Reviwer #1

Comments and Suggestions for Authors

Authors summarized the antibiotics commonly used in practice concerning CRE, CRAB and CR-Pseudomonas. 

They review the potential interest of cefiderocol, a new antibiotic, using siderophore with a novel mechanism of resistance using iron transport system.

They also, review the literature concerning the resistance described up to this date.

Some suggestions:

1) in the abstract:

the use of "and CR and 3rd Gen cephalosporin" is hard to read (including in throughout the manuscript). I would prefer using the term "carbapenem-producing Enterobacterales or ESBL-producers

Authors: We acknowledge the suggestion. The WHO uses this nomenclature that we followed. We know that beta-lactamases (carbapenemases and ESBLs) are a very common resistance mechanism, but carbapenem resistance may not be due only to the production of carbapenemases or, in the case of 3rd Gen cephalosporin, due to ESBL.

2) Introduction: Line 43 after citation [3]:

You may add "especially, the ones related to gram-negative bacteria"

Authors: Done.

3) Page 2 line 53 same suggestion as for the abstract concerning the term "CRE"

Authors: Previously answered.

4) Page 3 lines 110-130 : for me it is out of the scope of the article, at least it should be shortened. Indeed, you may need to open the discussion about the trend of a "carbapenem-sparing era", and the use / interest of new molecules including ceftazidim avibactam, and carbapenem + inhibitors, and why they do have interest right now.

Authors: Thank you for sharing your point of view. The authors just present the current therapeutic options, supported by the references. The idea is not discussing the new antibiotics (or associations) that are launched recently versus the “older” antibiotics. This is mentioned briefly and supported by references at the end of the paragraph with a reference for those that would like to know more. We saw this chapter as an introduction for the new molecule, Cefiderocol. The same type of introduction was done for each of the top priority microorganisms.

5) Page 7 Line 295 after citation [47]. You should add a word about the Kaye S. article in the NEJM which shows similar outcome for colistin monotherapy or in combination against CRAB (78%), in pneumonia (70%); 43% vs 37% mortality. https://evidence.nejm.org/doi/abs/10.1056/EVIDoa2200131

Authors: The suggested reference was added according to reviewer’s suggestion.

6) In the discussion page 10 , you may clarify the point that previously we were supposed to test Cefidericol using broth dilution for MIC determiniation with or without iron; currently EUCAST is calibrated using standard E-test manufactered by Shinogi.

Authors: We thank the reviewer for the suggestion. The following information was added to the discussion: “It should be taken in consideration that EUCAST firstly recommended that the antimicrobial susceptibility testing to cefiderocol could be performed by broth microdilution or by disk diffusion; in the former case, iron-depleted Mueller-Hinton broth should be employed [102]. However, due to limitations in MIC determination with commercially available tests, the current recommendations of EUCAST are the determination of the antimicrobial susceptibility to cefiderocol by disk diffusion (https://www.eucast.org/ast-of-bacteria/warnings).”

We would like to clarify that reference to an E-test manufactured by Shinogi was not added as we did not find this information available to the public.

7) Conclusion: I would definitely conclude towards cefiderocol insterest in comparison to new BL+inhibitors, in case other regimen are compromised; which is not clearly stated in the present manuscript.

Authors: Indeed, the association of cefiderocol with BL inhibitors might be a novel strategy to overcome this mechanism of resistance. There is a sentence in the conclusion referring to this possibility.

Reviewer 2 Report

1) What are the implications of these resistance mechanisms on clinical practice. Whilst the molecular and genetic aspects are well written, the implications of these mechanisms and their ongoing evolution is not clear for busy clinicans.

2) A more concise review of the current use of cefiderocol may be useful.

Author Response

We would like to express to the reviewers our sincere gratitude for the time and effort dedicated to reviewing the manuscript and your valuable and insightful and constructive comments to improve the overall quality of the manuscript.

Below you can find the answers to the reviewers.

Reviewer #2

Comments and Suggestions for Authors

1) What are the implications of these resistance mechanisms on clinical practice. Whilst the molecular and genetic aspects are well written, the implications of these mechanisms and their ongoing evolution is not clear for busy clinicans.

2) A more concise review of the current use of cefiderocol may be useful.

Authors: We acknowledge the reading of this manuscript and the comments of the reviewer. Indeed, we were more focused on the molecular aspects than clinical in the view of the scope of the Special Issue of the journal. Moreover, cefiderocol is still under clinical experience, and as we describe, there are conflicting clinical outcomes. Some show excellent activity, a few describe already development of resistance during therapy (these are very recent). It seems that P. aeruginosa and A. baumannii infections could be more problematic. Probably, this is why the reviewer would like to see a more concise review on current use of cefiderocol.

 First, the literature is not enough to draw clear conclusions. We find reports of excellent activity of cefiderocol, others report similar clinical outcomes compared with other antibiotic therapies, recent few reports describe development of resistance during therapy, and also, some strains can already be naturally resistant to cefiderocol (no previous exposure). All these points are discussed before the conclusion. We think that, right now, we cannot draw a simple conclusion. Instead, it was our objective to give an overview of the challenges of current and future use of cefiderocol. There are guidelines for its use and, based on the published cases, clinicians must be aware of potential limitations of the cefiderocol therapy. And this what authors would like to highlight (the conclusion).

Secondly, a more concise review of the current use of cefiderocol would be more useful for practitioner clinicians. Nevertheless, it would fit better in a journal with a more clinical scope.

Round 2

Reviewer 1 Report

Authors took into considering all my suggestions, and/or support the present form of the manuscript. I believe it is now ready for publication.